# Metagenomic next-generation sequencing to characterize potential etiologies of non-malarial fever in a cohort living in a high malaria burden area of Uganda

Lusajo Mwakibete[1☯], Saki Takahashi[2,3☯]*, Vida Ahyong[1], Allison Black[1], John Rek[4], Isaac Ssewanyana[4], Moses Kamya[4,5], Grant Dorsey[6], Prasanna Jagannathan[7,8], Isabel Rodríguez-Barraquer[2☯], Cristina M. Tato[1☯], Bryan Greenhouse[2☯]

1 Chan Zuckerberg Biohub, San Francisco, CA, United States of America, 2 Department of Medicine, Division of HIV, ID, and Global Medicine, EPPIcenter Research Program, University of California San Francisco, San Francisco, CA, United States of America, 3 Department of Epidemiology, Johns Hopkins Bloomberg School of Public Health, Baltimore, MD, United States of America, 4 Infectious Diseases Research Collaboration, Kampala, Uganda, 5 Department of Medicine, Makerere University, Kampala, Uganda, 6 Department of Medicine, Division of HIV, ID, and Global Medicine, University of California San Francisco, San Francisco, CA, United States of America, 7 Department of Medicine, Stanford University, Palo Alto, CA, United States of America, 8 Department of Microbiology and Immunology, Stanford University, Palo Alto, CA, United States of America

☯ These authors contributed equally to this work.
* saki.takahashi@jhu.edu

**Data Availability Statement:** The SARS-CoV-2, Influenza A, and RSV genomes generated in this

## Abstract

Causes of non-malarial fevers in sub-Saharan Africa remain understudied. We hypothesized that metagenomic next-generation sequencing (mNGS), which allows for broad genomic-level detection of infectious agents in a biological sample, can systematically identify potential causes of non-malarial fevers. The 212 participants in this study were of all ages and were enrolled in a longitudinal malaria cohort in eastern Uganda. Between December 2020 and August 2021, respiratory swabs and plasma samples were collected at 313 study visits where participants presented with fever and were negative for malaria by microscopy. Samples were analyzed using CZ ID, a web-based platform for microbial detection in mNGS data. Overall, viral pathogens were detected at 123 of 313 visits (39%). SARS-CoV-2 was detected at 11 visits, from which full viral genomes were recovered from nine. Other prevalent viruses included Influenza A (14 visits), RSV (12 visits), and three of the four strains of seasonal coronaviruses (6 visits). Notably, 11 influenza cases occurred between May and July 2021, coinciding with when the Delta variant of SARS-CoV-2 was circulating in this population. The primary limitation of this study is that we were unable to estimate the contribution of bacterial microbes to non-malarial fevers, due to the difficulty of distinguishing bacterial microbes that were pathogenic from those that were commensal or contaminants. These results revealed the co-circulation of multiple viral pathogens likely associated with fever in the cohort during this time period. This study illustrates the utility of mNGS in elucidating the multiple potential causes of non-malarial febrile illness. A better understanding of the pathogen landscape in different

analysis were deposited in their respective GISAID databases (see S2 Table for accession numbers). The R code used to produce Fig 2, Nextstrain trees generated in this analysis, and corresponding Snakefiles are available on GitHub at: https://github.com/czbiohub/rapid-response-prism. The SRA files of non-host reads associated with this analysis are available at: https://www.ncbi.nlm.nih.gov/bioproject/PRJNA870959. CZ ID public links to heatmaps visualizing the mNGS detections are available at: https://czid.org/pub/7pxmXbpFTL. Note that while the two batches were separately processed for the analysis, for simplicity they have been combined in this public link. The relevant clinical and demographic data are available in S1 File.

**Funding:** We acknowledge sources of funding support, including from the Schmidt Science Fellows, in partnership with the Rhodes Trust (ST); Chan Zuckerberg Biohub Investigator program (IRB, BG); NIH/NIAID U19AI089674 (JR, IS, MK, GD, PJ, IRB, BG); NIH/NIAID U01AI150741 (ST, PJ, IRB, BG). The funders had no role in study design, data collection and analysis, decision to publish, or preparation of the manuscript.

**Competing interests:** The authors have declared that no competing interests exist.

settings and age groups could aid in informing diagnostics, case management, and public health surveillance systems.

## Introduction

Fever is a frequent disease manifestation and a common reason for seeking health care services in resource-limited settings. Fever has many underlying etiologies, including infection with a broad array of pathogens such as malaria caused by the *Plasmodium falciparum* protozoan parasite, as well as other parasites, viruses, and bacteria. Given the non-specific disease presentation, determining etiologies of fever based solely on clinical examination is difficult, and therefore diagnostic testing is needed. For malaria, a major cause of morbidity and mortality in Africa [1], recent increases in point-of-care and laboratory diagnostic testing have led to demonstrable improvements in malaria case management [2]. However, there are sustained knowledge gaps on fever etiologies that do not result in a malaria diagnosis due to the lack of testing availability for other pathogens, and the relative burden of other emerging pathogens of global importance (e.g., arboviruses) remains largely uncharacterized. The acquisition of clinical immunity to *P. falciparum* further confounds the study of fever etiologies in malaria-endemic settings. Individuals acquire clinical immunity against malaria over repeated infections, leading to the ability to tolerate malaria parasites in the blood without developing fever [3]. In individuals with parasitemia who have acquired high levels of clinical immunity, *P. falciparum* is unlikely to be the cause of febrile illness. Taken together, the limited availability of diagnostics for pathogens beyond malaria results in many undiagnosed illnesses [4], missed opportunities for targeted treatments [5], unnecessary empiric use of antibiotics [6], and public health surveillance systems that provide an incomplete picture of the pathogen landscape [7].

Direct pathogen detection platforms such as metagenomic next-generation sequencing (mNGS) can be harnessed to meet this multifaceted challenge of identifying non-malarial fever etiologies. mNGS allows for the broad and unbiased genomic-level detection of infectious agents in a biological sample. This powerful technology has played a key role in disparate fields including clinical diagnostics [8], microbiome characterization [9], outbreak detection [10,11], and transcriptomics [12]. As highlighted in a recent review piece by Ko and colleagues [13], multiplexed pathogen testing can be particularly advantageous in resource-constrained settings, as a single assay can generate information about a suite of microbes in a sample. Recent mNGS studies in Uganda [14], Kenya [15], and Cambodia [11] have revealed new insights into the local patterns of infectious etiologies of non-specific disease presentations including fever and pneumonia. As recently underscored by the SARS-CoV-2 pandemic, unbiased disease surveillance can also provide insight with regard to incidence of newly emerging infections, especially in geographies where implementation of widespread, disease-specific diagnostics is constrained by competing demands on resources [16]. In addition to identifying the microbial composition of a sample, a key strength of the mNGS approach is the possibility to generate partial or whole pathogen genomes for downstream epidemiologic and genomic investigations. There are various considerations for selecting the optimal nucleic acid to evaluate in metagenomic studies [17,18]. RNA libraries are required for capturing RNA viruses, and there is a substantially higher signal than DNA from actively replicating microbes due to the amplification of microbial RNA, especially when combined with host RNA depletion methods. Likewise, selecting DNA for a metagenomic study is important when the RNA quality is low, when identifying latent viruses, or when antibiotics have been administered, which would prevent detection of actively replicating bacteria.

Here, we performed RNA mNGS as a research tool to systematically study the potential causes of non-malarial febrile illness within a well-characterized, multi-generational, representative cohort of individuals living in an area with high malaria burden in eastern Uganda. Given the timing of this study, which ran from December 2020 to August 2021, we were also uniquely situated to detect SARS-CoV-2 infections and the co-circulation of multiple respiratory viral pathogens during this time period. Given the difficulty of discriminating bacterial microbes that are pathogenic from those that are commensal or contaminants, for the purposes of this study we focused on the detections of viruses and *P. falciparum* malaria by mNGS.

## Methods

### Study population and design

This pilot study was nested within the ongoing Program for Resistance, Immunology, Surveillance, and Modeling of Malaria in Uganda (PRISM) Border Cohort study in the Tororo and Busia districts of Uganda. The design and population of this cohort study have recently been described elsewhere [19]. Briefly, all households in the parishes of Osukuru (Tororo district), Kayoro (Tororo district), and Buteba (Busia district) were enumerated to generate a sampling frame to recruit households into the study. In August 2020, households were randomly selected and screened for eligibility to participate. Inclusion criteria for a household to participate included having at least two members aged 5 years or younger. All permanent members of an enrolled household who met eligibility criteria were screened for enrollment. The cohort was dynamic, so any permanent members that joined an enrolled household were screened for enrollment.

The approximately 500 participants from the 80 households enrolled in the cohort were encouraged to come to a dedicated study clinic open 7 days a week for all of their medical care free of charge and were reimbursed for their transport costs, minimizing barriers to accessing care. Routine study visits were conducted every 4 weeks, and included a standardized evaluation and blood collection by finger prick or heel stick (if < 6 months of age) or by venipuncture (if 6 months of age and older). At each study visit where participants reported a fever or history of fever in the last 24 hours, testing for asexual malaria parasites was performed by microscopic examination of a thick blood smear. Participants were diagnosed with malaria if positive and managed according to national guidelines [20]; the incidence of malaria in the cohort was approximately 1 to 3 episodes per person per year in children, and lower in adults [19]. In addition, a sample was collected for subsequent testing via ultrasensitive qPCR for *Plasmodium falciparum* malaria [21].

Participants were eligible for this pilot mNGS study if they had a fever (determined as either having a fever at the time of visit with objective temperature > 38˚C, or having self-reported fever in the past 24 hours) or a rash, accompanied by a negative malaria blood smear result. Participants of all ages were included. Individuals meeting these criteria, but who were positive for malaria by qPCR, were included. Our expectation was that submicroscopic parasitemia would be prevalent in this setting and unlikely to be the source of the febrile illness. For each participant visit that met these criteria, we obtained a combined oropharyngeal/mid-turbinate swab and 200 μL of plasma, each collected directly into separate cryovials containing DNA/RNA Shield transport and storage media, and frozen at -80˚C. For a minority of visits, only one of the sample types was collected.

### Ethics statement

The study protocol was reviewed and approved by the Makerere University School of Medicine Research and Ethics Committee, the Uganda National Council of Science and

Technology, the University of California, San Francisco, Human Research Protection Program, the London School of Hygiene & Tropical Medicine Ethics Committee, and the Stanford University Research Compliance Office. Written informed consent was obtained for adults (18 years of age or older) and the parent/guardian of each participant under 18 years of age prior to enrolment into the study. Informed assent was obtained for children 8–17 years of age.

## mNGS sample processing and sequencing

For each collected sample, nucleic acid was extracted using the quick-DNA/RNA Pathogen MagBead kit (Zymo Research). Extracted nucleic acid was treated with DNAse to isolate RNA alone and run on a TapeStation for quality control to examine RNA integrity. Water controls were used to characterize background contamination, as well as 25 pg of a positive control spike-in (RNA standard dilution series from External RNA Controls Consortium (ERCC)). Plate maps were designed to detect and minimize cross-contamination between wells by interspersing samples and water controls. Samples were run in two experimental batches (see **S1 Table**). FastSelect -rRNA HMR (Qiagen) was used for human RNA ribosomal depletion, which is known to knockdown ~98% of human ribosomal RNA targeting the cytoplasmic (5S,5.8S,18S,28S) and mitochondrial (12S,16S) rRNAs, and specifically targets human, mouse, rat, and other mammalian species [22]. RNA was reverse-transcribed to attain cDNA, which was used to construct and barcode sequencing libraries using the NEBNext Ultra II Library Prep Kit (New England Biolabs). The RNA sequencing libraries underwent 150-nt paired-end Illumina sequencing. The target reads were at least 5 million reads per sample to attain enough coverage depth per respective sample.

## mNGS bioinformatic analysis using CZ ID

We used the CZ ID bioinformatics pipeline v6.8 (http://czid.org), an established, open-source sequencing analysis platform for raw mNGS data which enables the detection and taxonomic identification of microbes [23]. Briefly, the pipeline filters out reads mapping to the human host (using *STAR [24]*) and removes low-quality (using *Price Seq [25]*), low-complexity (*LZW*), and duplicate reads as well as adapter sequences (using *Trimmomatic [26]*). The final composition of reads in each sample was determined by querying the remaining reads on NCBI's nucleotide (NT) and non-redundant protein (NR) databases using the *GSNAP-L [27]* and *RAPSearch2 [28]* programs, respectively.

For each sample, significant microbial detections were determined from the unique reads per million (rPM) that mapped to specific microbial taxa, genera, and species. To do so, we applied the following four threshold filters to determine the presence of a microbe in a sample: nucleotide (NT) Z-score $\geq$ 1 (calculated from the mass-normalized background model created on CZ ID using the water controls), NT rPM $\geq$ 10 (aiming for a minimum of at least 10 reads of a mapping to a certain microbe per 1 million reads), non-redundant protein (NR) rPM $\geq$ 5, and average NT alignment $\geq$ 50 base pairs. We chose to apply both the NT and NR sequence filter to minimize potential false positive detections from mis-alignment or mis-annotation in the NCBI database. Furthermore, the benchmarking results of CZ ID's precision/recall on known data sets yielded high AUPR values, indicating the detection of known organisms at higher abundance than any false positives due to mis-alignments or mis-annotations (https://chanzuckerberg.zendesk.com/hc/en-us/articles/4418861817748-CZ-ID-Pipeline-Update-Making-our-Pipeline-More-Scalable). We conservatively excluded the bacterial reads in samples with low input ($<$ 25 pg) because of the potential amplification of background contaminants from reagents and/or the environment (see **S1 Fig**). For two swab samples with ERCC

spike-in counts of 0, Z-scores were not calculable since the normalization method relies on ERCC counts, so only the other three threshold filters were applied.

## Alignment pipelines for generating consensus viral genomes

**SARS-CoV-2.** Consensus viral genomes were obtained using CZ ID's consensus genome pipeline v3.4.7. Non-host reads from each of the 11 samples in which SARS-CoV-2 was detected by CZ ID were aligned to the SARS-CoV-2 Wuhan-Hu-1 reference genome (MN908947.3) using *minimap2* [29]. Aligned reads were trimmed using *trim galore* [30] and adapters, low-quality reads (Phred quality score < 20), and short sequences (< 20 base pairs) were removed. Trimmed reads were then again aligned to MN908947.3 using *minimap2*, primers were trimmed using *iVar* v1.3.1 [31,32], and consensus genomes were generated using *iVar consensus*. Bases were called if they had a depth of $\geq$ 10 reads. Bases that were not called were identified as N, and SNPs were called using *samtools* v1.9 and *bcftools* v1.9. In total, 9 SARS-CoV-2 consensus genomes were obtained with $\geq$ 90% coverage breadth.

**Influenza A virus.** FASTQ files of the reads mapping to Influenza A from the 14 samples in which Influenza A was detected by CZ ID were downloaded. As all of the Influenza A reads were of the H3N2 subtype, reads were mapped to the Influenza A H3N2 reference genome from NCBI (GenBank Accession: 1559708) [33] using Geneious Prime 2022.1.1. The software was used to map reads and to obtain consensus genomes. Geneious was used rather than the CZ ID consensus genome pipeline (as we did for SARS-CoV-2 and RSV) because the Influenza A virus genome has multiple segments, thus requiring separate alignments to each segment for every sample. Nucleotide base calling required at least 90% similarity across the reads per respective position, and bases were called if they had a depth of $\geq$ 10 reads. In total, 9 Influenza A consensus HA genes were obtained with $\geq$ 80% coverage breadth.

**Respiratory syncytial virus (RSV).** Similarly as for SARS-CoV-2, CZ ID's consensus genome pipeline v3.4.7 was used to obtain consensus viral genomes from the 12 samples in which RSV was detected by CZ ID (11 samples were of the RSV-A subtype and 1 sample was of the RSV-B subtype). The very low read count of the latter, the sole plasma sample and with only 19 NT reads, precluded generation of a consensus genome. Non-host reads from each sample were aligned to their closest RSV-A reference genome based on genetic similarity and coverage depth. These reference genomes varied between samples; the ones used were NCBI Accession: MN306017.1, KY967363.1, KY654513.1 and KC731482.1. As before, aligned reads were trimmed using *trim galore* and adapters, low-quality reads (Phred quality score < 20), and short sequences (< 20 base pairs) were removed. Bases were called if they had a depth of $\geq$ 5 reads, the default setting on the CZ ID pipeline. Bases that were not called were identified as N, and SNPs were called using *samtools* and *bcftools*. In total, 9 RSV consensus genomes were obtained with $\geq$ 90% coverage breadth.

## Reconstructing viral phylogenies

The Nextstrain platform [34] was used to perform phylogenetic inference of the SARS-CoV-2, Influenza A, and RSV genomes obtained from this analysis. To ensure proper contextualization of the sequences generated as part of this study, we sourced publicly available sequence data from relevant GISAID databases, as described below.

To analyze SARS-CoV-2 sequences, we downloaded all SARS-CoV-2 genome sequences available from the GISAID EpiCoV database [35]. Within our Nextstrain build, we specified that all sequences sampled from Uganda available on GISAID should be included in the phylogenetic analysis. Together with the SARS-CoV-2 genomes that we sequenced as part of this study, these Ugandan genomes formed our focal set. Contextual sequences from other

countries were included in the analysis based on their genetic similarity to sequences in the focal set, with more genetically similar contextual sequences prioritized for inclusion in the analysis dataset. To retain the historical diversity of SARS-CoV-2 in our analysis, and to ensure accuracy of the molecular clock, an equitable number of contextual sequences were sampled per month across the entire duration of the pandemic. The final dataset used for phylogenetic analysis included 2,686 SARS-CoV-2 genomes sampled from December 2019 to November 2021, of which 687 were sampled from Uganda (including the 9 generated in this study).

To analyze Influenza A sequences, we downloaded all 9,193 Influenza A virus H3N2 HA sequences available on the GISAID EpiFlu database [36] from January 2020 to March 2022. Within our Nextstrain build, we specified that all sequences sampled from Africa available from EpiFlu should be included in the phylogenetic analysis, along with the Influenza A viruses sequenced as part of this study. Contextual sequences from other countries were included in the analysis; sub-sampling of contextual sequences by country, year and month was used to ensure global representation and accuracy of the molecular clock. The final dataset used for phylogenetic analysis consisted of 3,825 Influenza A HA sequences (including the 9 generated in this study).

To analyze RSV sequences, we downloaded all 677 RSV genomes available on the GISAID EpiRSV database [35] from January 2020 to February 2022. Within our Nextstrain build, we specified that all sequences sampled from Africa available from EpiRSV should be included in the phylogenetic analysis, along with the RSV viruses sequenced as part of this study. Contextual sequences from other countries were included in the analysis; sub-sampling of contextual sequences by country, year and month was used to ensure accuracy of the molecular clock. The final dataset used for phylogenetic analysis consisted of 469 RSV genomes (including the 9 generated in this study).

We used Nextstrain Augur to perform a multiple sequence alignment of input sequences using *MAFFT [37]* and to strip the multiple sequence alignment to the reference genome, which removes sequence insertions that introduce gaps in the reference sequence. *IQ-TREE [38]* was used to generate a maximum likelihood genetic divergence tree. We generated temporally resolved trees using *TreeTime [39]*. Nextstrain phylogenetic trees were exported as JSON files by the pipeline, which we then visualized and explored in the web browser using Nextstrain Auspice. Figures were generated from SVG files of these visualized trees. All analysis was conducted using the R statistical software versions 4.0.2 and 4.1.3.

## Data availability statement

The SARS-CoV-2, Influenza A, and RSV genomes generated in this analysis were deposited in their respective GISAID databases (see **S2 Table** for accession numbers). The R code used to produce Fig 2, Nextstrain trees generated in this analysis, and corresponding Snakefiles are available on GitHub at: https://github.com/czbiohub/rapid-response-prism. The SRA files of non-host reads associated with this analysis are available at: https://www.ncbi.nlm.nih.gov/bioproject/PRJNA870959. CZ ID public links to heatmaps visualizing the mNGS detections are available at: https://czid.org/pub/7pxmXbpFTL. Note that while the two batches were separately processed for the analysis, for simplicity they have been combined in this public link. The relevant clinical and demographic data are available in **S1 File**.

## Inclusivity in global research

Additional information regarding the ethical, cultural, and scientific considerations specific to inclusivity in global research is included in **S1 Checklist**.

## Results

### Clinical, demographic, and geographic characteristics of participants

Samples were collected for this study between December 2020 and August 2021. Of the 624 total febrile study visits in the cohort during the time period of this study, 357 had a negative test result for malaria parasites by blood smear. We were able to evaluate a large majority of these non-malarial febrile illnesses in this representative cohort. In addition, samples were obtained from 3 afebrile participants who had a rash, which was the other eligibility criteria for this study (**S2 Fig**). The characteristics of the study participants in this mNGS study are provided in **Table 1** and **Fig 1**. Overall, samples from 212 individuals from 80 households were included in this study, of whom 50% were < 5 years of age, 20% were between 5 and 15 years

**Table 1. Demographic and clinical characteristics of the study participants.**

| Individuals (N = 212) | |
|---|---|
| Age at cohort study enrollment | < 5 years: 105 (49.5%)<br>5–15 years: 44 (20.8%)<br>16 years or older: 63 (29.7%) |
| Sex | Female: 128 (60.4%)<br>Male: 84 (39.6%) |
| District | Tororo: 159 (75.0%)<br>Busia: 53 (25.0%) |
| Number of visits with mNGS sample collection | 1: 149 (70.3%)<br>2: 36 (17.0%)<br>3: 20 (9.4%)<br>4: 3 (1.4%)<br>5: 4 (1.9%) |
| Number of unique households | 80 |
| **Study visits with sample collection (N = 313)** | |
| Date range | December 14, 2020—August 22, 2021 |
| Malaria blood smear result | Positive: 1 (0.3%)<br>Negative: 309 (98.7%)<br>Not done*: 3 (1.0%) |
| *Plasmodium falciparum* malaria qPCR result | Positive: 74 (23.6%)<br>Negative: 238 (76.0%)<br>Not done: 1 (0.3%) |
| Objective temperature at study visit | < 37.5˚C: 215 (68.7%)<br>37.5–38.0˚C: 47 (15.0%)<br>≥ 38.0˚C: 51 (16.3%) |
| Symptom reported: fever | Yes: 310 (99.0%)<br>No: 3 (1.0%)* |
| Symptom reported: cough | Yes: 187 (59.7%)<br>No: 126 (40.3%) |
| Symptom reported: headache | Yes: 76 (24.3%)<br>No: 213 (68.1%)<br>Cannot assess: 24 (7.7%) |
| Symptom reported: fatigue | Yes: 36 (11.5%)<br>No: 267 (85.3%)<br>Cannot assess: 10 (3.2%) |
| mNGS specimens collected at study visit and included in final analysis** | Swab & plasma: 273 (87.2%)<br>Swab only: 21 (6.7%)<br>Plasma only: 19 (6.1%) |

The top 4 most prevalent reported symptoms are listed. A "cannot assess" designation could have been given for symptom reporting in young children. *These participants had a rash and no fever. **Three additional swabs and two additional plasma specimens were collected but did not pass QC.

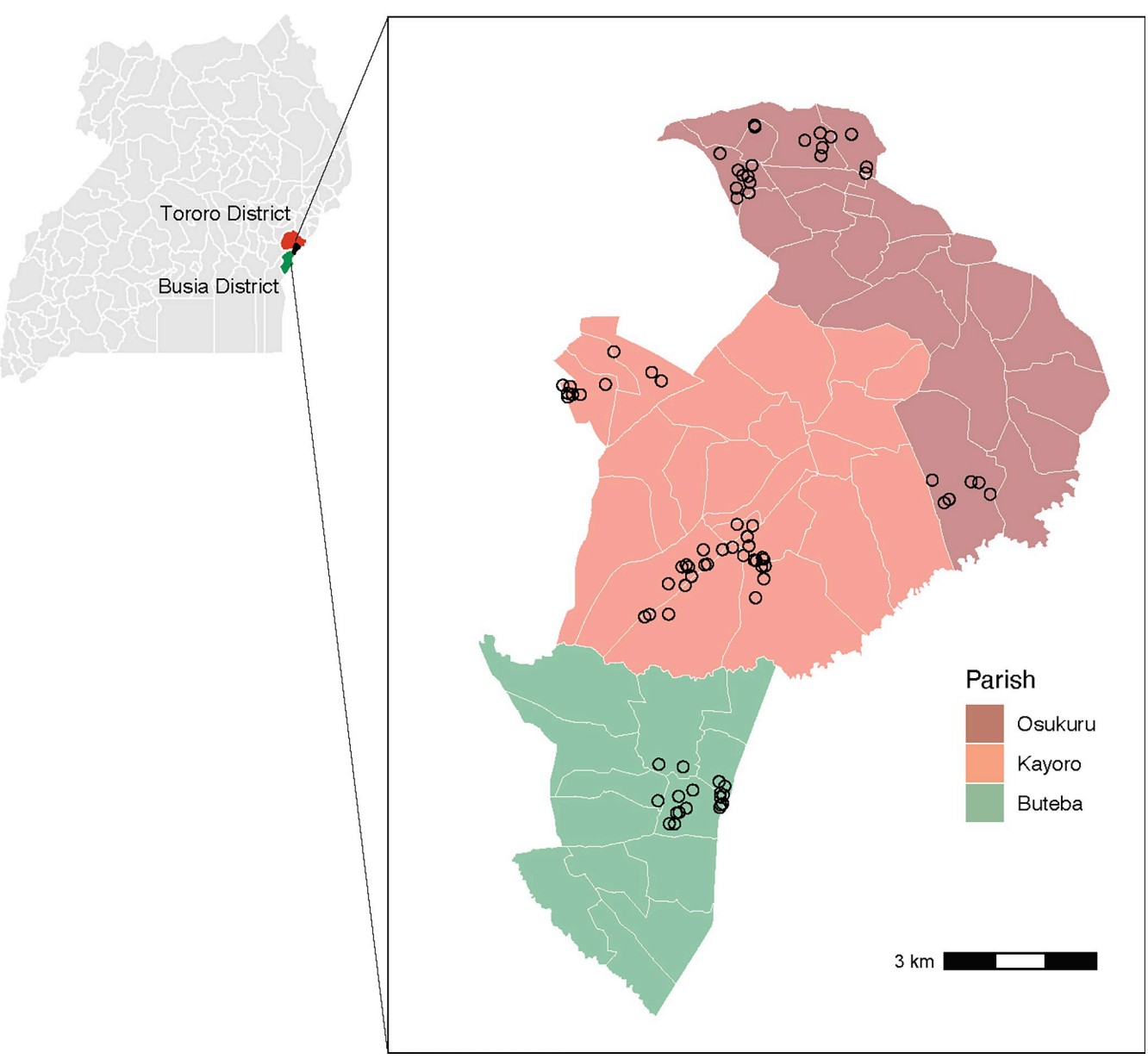

**Fig 1. Maps of the study area.** Map of the 136 districts in Uganda, highlighting the two districts included in the PRISM Border Cohort study (Tororo District in red, Busia District in green). The black area represents the 3 parishes from which households were enrolled. The outlines in white reflect district borders. The base layer shapefile was obtained from the Operational Data Portal (https://data.unhcr.org/en/documents/details/83043), and is licensed under a Creative Commons Attribution 3.0 International License (CC BY 3.0). **Inset:** Map of the PRISM Border Cohort study area. Each point represents 1 household enrolled in the study that had mNGS sample collection performed (80 households in total). Osukuru and Kayoro are 2 of the 88 parishes in Tororo District; Buteba is one of the 63 parishes in Busia District. The outlines in white reflect village borders. The base layer shapefile was obtained from the Uganda Bureau of Statistics and downloaded through humdata.org, and is licensed under a Creative Commons Attribution for Intergovernmental Organisations (CC BY IGO).

of age, and 30% were 16 years of age or older. 70% of individuals had sample collection at one visit, and the rest had sample collection at multiple visits (up to 5).

Sample collection was performed at a total of 313 study visits. Even though participants had to be parasite negative by blood smear to be eligible for this mNGS study and have a sample collected, at a quarter of visits (74/313), participants were subsequently found to be positive for *Plasmodium falciparum* by ultrasensitive qPCR. The most frequently reported symptoms, in

addition to fever, were cough, headache, and fatigue. 294 swab samples and 292 plasma samples collected for this study passed CZ ID's QC filters and were included in this analysis (3 swab samples and 2 plasma specimens were collected but did not pass QC), and most were paired from the same visit.

## Microbes detected by mNGS at visits with malaria blood smear-negative fever episodes

We applied stringent threshold filters for detecting microbes in order to minimize the effects of noise introduced by low sample input and background contaminants. Bacterial, viral, and eukaryotic microbes were detected at 92%, 47%, and 63% of the 313 visits, respectively (**Table 2**). We further examined 18 detected viruses that are known to be pathogenic to humans and have previously been causally linked to febrile disease, focusing on acute respiratory and gastrointestinal (GI) viral infections (**S3 Table**). Overall, 119 of the 313 visits (38%) had at least 1 respiratory viral pathogen detected, 5 visits had a GI viral pathogen detected, and 4 visits had more than 1 viral pathogen detected. The proportion of study visits with at least 1 viral pathogen detected decreased by age: 49% in participants < 5 years of age, 30% in participants 5–15 years of age, and 28% in participants 16 years of age or older. As the likelihood of viral pathogen detection depends on the sample type(s) tested, results at the 273 visits with paired sample collection are presented in Table 2.

Of the 273 visits with plasma-swab pairs tested by mNGS, 62 were previously characterized as positive for *Plasmodium falciparum* malaria by qPCR in the cohort study, of which 22 were also positive by mNGS. *Plasmodium falciparum* malaria was detected by mNGS in an additional 35 visits, including 1 visit in which a qPCR result was not available (**Tables 3** and S4). Overall, 73% of *Plasmodium falciparum* malaria results were in agreement between qPCR and mNGS. In addition, we found a positive relationship between parasite density values obtained from qPCR and NT rPMs mapping to *Plasmodium falciparum* by mNGS among the 40 plasma

**Table 2. mNGS detection of microbes and viral pathogens at study visits.**

| Microbes detected at visit | At all study visits N = 313 visits | At visits with paired swab and plasma N = 273 visits |
|---|---|---|
| No microbes | 10 | 2 |
| Bacteria only | 44 | 40 |
| Virus only | 3 | 1 |
| Eukaryota only | 8 | 1 |
| Bacteria + virus | 59 | 46 |
| Bacteria + eukaryota | 104 | 102 |
| Virus + eukaryota | 4 | 1 |
| Bacteria + virus + eukaryota | 81 | 80 |
| **Viral pathogens detected at visit** | **At all study visits N = 313 visits** | **At visits with paired swab and plasma N = 273 visits** |
| No respiratory viral pathogens | 194 | 171 |
| 1 respiratory viral pathogen | 116 | 99 |
| 2 respiratory viral pathogens | 2 | 2 |
| 3 respiratory viral pathogens | 1 | 1 |
| No GI viral pathogen | 308 | 268 |
| 1 GI viral pathogen | 5* | 5* |

Refer to S3 Table for the list of viral pathogens considered. *A co-detection with a respiratory viral pathogen was detected at 1 visit.

**Table 3. Frequency table of detection of *P. falciparum* malaria by mNGS and qPCR at the 273 study visits with paired mNGS sample collection, and co-detection with viral pathogens.**

| N = 273 visits with paired mNGS sample collection | Malaria qPCR (-) N = 210 visits | Malaria qPCR (+) N = 62 visits | No malaria qPCR available N = 1 visit |
|---|---|---|---|
| **Malaria mNGS (-) in plasma N = 216 visits** | 176 visits 71 visits had at least 1 viral pathogen | 40 visits 12 visits had at least 1 viral pathogen | 0 visits |
| **Malaria mNGS (+) in plasma N = 57 visits** | 34 visits 15 visits had at least 1 viral pathogen | 22 visits 8 visits had at least 1 viral pathogen | 1 visit 0 visits had at least 1 viral pathogen |

mNGS results for malaria are from the plasma specimen only. The co-detection of respiratory/gastrointestinal viral pathogens (see S3 Table) in either the plasma or swab specimen are tabulated in the bullet points. The list and frequency of viruses detected are tabulated in S8 Table.

samples with non-zero values on both assays (**S3 Fig**). All samples with greater than 13.4 parasites per μL by qPCR were positive for *Plasmodium falciparum* by mNGS. 19 of the 34 visits that were qPCR negative and mNGS positive had at least 1 positive blood smear or qPCR result within the 3 months before or after the visit, indicating participants may have harbored parasites below the limit of detection by qPCR.

Interestingly, reads mapping to *Plasmodium ovale* were detected in five plasma samples. Of these five samples, three had the majority of *Plasmodium* reads mapping specifically to the *P. ovale* species. While *P. ovale* infection is not systematically tested for in this cohort (as *P. falciparum* is the dominant human malaria pathogen in Uganda), *P. ovale* has previously been detected in other investigations in this setting [40].

## Microbial landscapes across samples

From the total of 586 samples, 735 unique microbial species were detected (468 bacteria, 238 eukaryota, 42 viruses, and 3 archaea) (**Fig 2**). The most commonly detected microbial species in plasma samples after filtering were *Malassezia restricta* (60 samples, 21%) and *Plasmodium falciparum* (57 samples, 20%). 79 plasma samples had zero microbes detected at the species level after applying threshold filters (**S4 Fig**; the bacterial genera detected in plasma samples are tabulated in **S5 Table**). The most commonly detected microbial species in swab samples after filtering were *Dolosigranulum pigrum* (152 samples, 52%) and *Moraxella catarrhalis* (141 samples, 48%). 2 swab samples had zero microbes detected at the species level after applying threshold filters (**S5 Fig**; the bacterial genera detected in swab samples are tabulated in **S6 Table**). Of the 42 viral species detected across all samples, 26 were human viruses (including the 18 known pathogenic human viruses described above).

For the remainder of this analysis, we focused on the viruses identified in these samples for which we had the highest confidence that detection was likely to be associated with illness. The rationale for not further characterizing the bacterial reads here is two-fold. The significantly lower sample inputs in plasma samples (i.e., multiple orders of magnitude below swab samples) makes it difficult to distinguish a true bacterial detection from a false positive or contaminant. Consistent with many of the detected bacteria being commensals, the most common species were expected skin or mucosa flora in healthy individuals. However, some of these may have been associated with illness. In plasma samples, the mNGS pipeline identified three cases of *Mycobacterium* and one case of *Rickettsia* (all with genus-level confidence only), as well as two cases of *Pseudomonas aeruginosa* and one case of *Streptococcus mitis*.

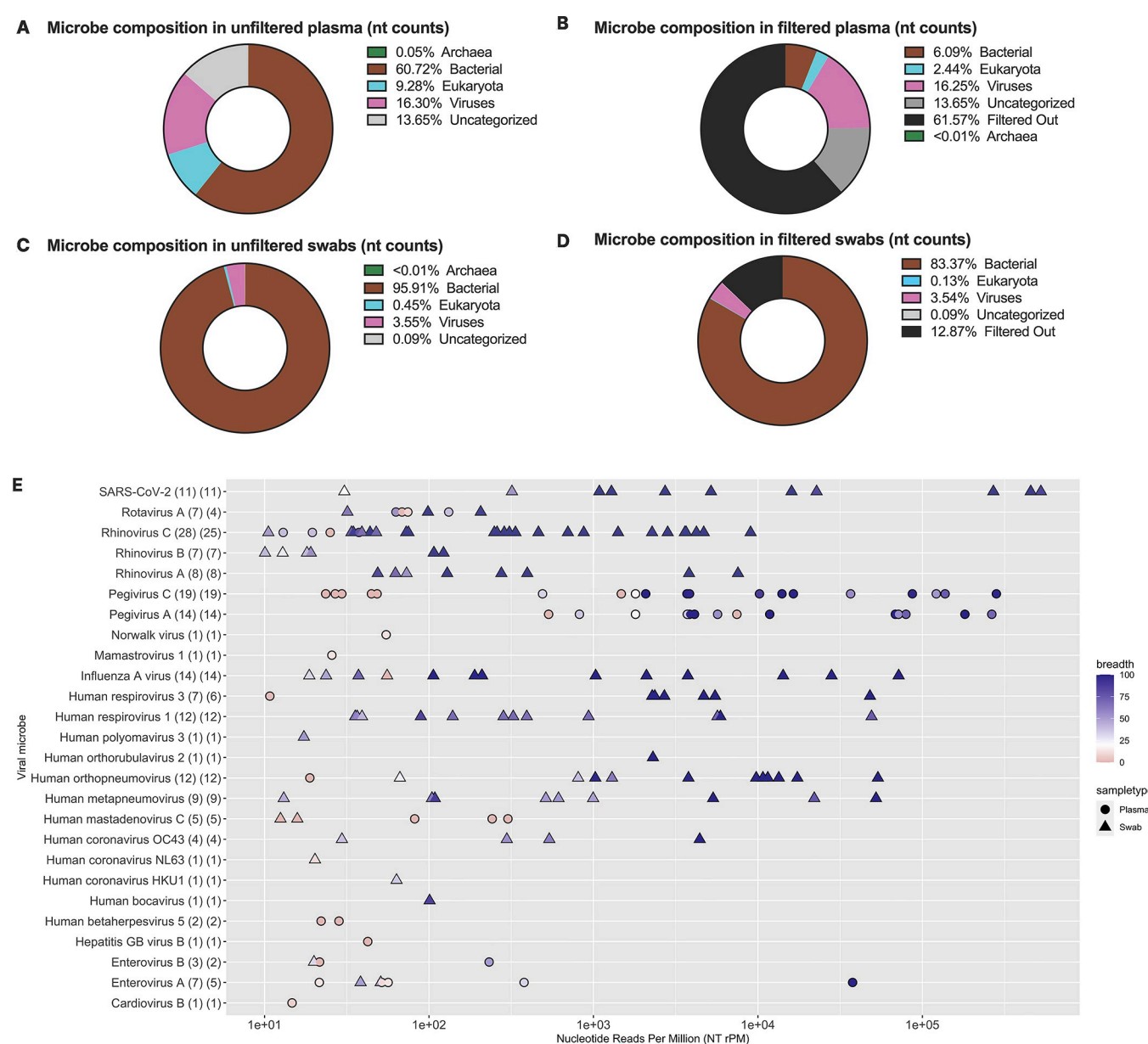

**Fig 2. Overall microbial composition in plasma and swab samples by kingdom, and species-level detection of viral microbes. (A,C)** Pre-filtering and **(B,D)** post-filtering microbial composition in plasma and swabs by species-level total nucleotide read counts across all samples, stratified by archaea, bacteria, eukaryota, viruses, and uncategorized. Uncategorized reads include vectors, uncultured microorganisms, uncultured prokaryotes, unidentified soil organisms, otherwise unidentified organisms, and taxa with neither family nor genus classification. Filters applied: NT Z-score $\geq$ 1, NT rPM $\geq$ 10, NR rPM $\geq$ 5, and average NT alignment $\geq$ 50 base pairs. The bacterial reads in all samples with low sample input (i.e., $<$ 25 pg) were also excluded. **(E)** Species-level human viral microbes detected (y-axis) with their NT rPM (x-axis). The point color denotes the coverage breadth of the particular sample and shape denotes sample type. On the y-axis label, the first value in parenthesis represents the number of unique samples with that viral microbe detected. The second value in parenthesis represents the number of unique participant visits where that viral microbe was detected.

**Fig 2E** summarizes the frequency, normalized NT read count, and breadth of genome recovered for each human viral microbe detected at the species level, in both sample types. The three most commonly detected virus species in plasma samples were Pegivirus C (19 detections), Pegivirus A (14 detections), and Enterovirus A (5 detections). Viral co-detection was observed in 15 plasma samples (11 co-detections of Pegivirus A and C; 2 co-detections of

Human mastadenovirus C and Pegivirus C; 1 co-detection of Enterovirus A and Human beta-herpesvirus 5; 1 co-detection of Hepatitis GB virus B and Pegivirus A and C). The three most commonly detected virus species in swab samples were Rhinovirus C (25 detections), Influenza A virus (14 detections), and Human respirovirus 1 (12 detections). Viral co-detection was observed in four swab samples (Rhinovirus A and B; SARS-CoV-2 and Human respirovirus 1; Human respirovirus 3 and Rhinovirus C; Rhinovirus C and Rotavirus A). Across the viral species detected, we observed increasing coverage breadth with greater normalized NT read counts.

As expected, concordance between viral microbes detected in plasma vs. swab samples was low (**S6 Fig**). Of the 120 sample pairs in which at least one human virus was detected, we found that 81 pairs had viral microbes in only the swab sample and 19 pairs had viral microbes in only the plasma sample. Among the remaining 20 pairs in which viral microbes were detected in both, the compositions for 8 pairs were identical between the sample types and 12 pairs were discordant. While as expected, respiratory viruses were primarily detected in respiratory swab samples, there were a few instances where respiratory and gastrointestinal viruses were also (or only) detected in plasma samples.

## Epidemiology of acute viral pathogens in the cohort study

Collapsing over specimen types, the most prevalent acute respiratory viral pathogens identified in this cohort were rhinovirus (40 detections), Influenza A (14 detections, all of the H3N2 subtype), parainfluenza (19 detections), RSV (12 detections), and SARS-CoV-2 (11 detections) (**Fig 3B–3C**). We first analyzed the temporal distribution of the acute respiratory and GI viral pathogens that were detected by mNGS. Most (9 of 11) of the SARS-CoV-2 cases were detected between May to July 2021, which coincided with when the Delta variant of SARS-CoV-2 was circulating in this population. Interestingly, we found that many other respiratory viruses, notably Influenza A, were also co-circulating in the cohort during this time, consistent with national-level case reports of SARS-CoV-2 and Influenza A (**Fig 3A**). The frequency and composition of respiratory viral pathogens detected varied considerably by age (**Fig 3D**). For example, the majority of rhinovirus, Influenza A, parainfluenza, and RSV detections were in children < 5 years of age. In contrast, the majority of SARS-CoV-2 detections were in individuals 16 years of age or older. While the estimated prevalence of most pathogens among study participants decreased by age, notable exceptions to this trend included SARS-CoV-2, seasonal coronavirus, and adenovirus (**S7 Fig**).

Clinical presentations varied by pathogen group and age (**Fig 4**). While there was considerable heterogeneity within and between pathogen groups, and small sample sizes prevented formal testing for multivariate associations, a few patterns emerged: Influenza A infection was associated with high objective temperatures (9 of 14 visits with $\geq 38.0°C$), cough was reported across the pathogen groups, and the prevalence of headache was highest among participants with SARS-CoV-2 (8 of 11 participants). We also found evidence of clustering of infections at the household level. In 11 of the 80 households, multiple individuals were infected with the same respiratory pathogen during the sampling period (**S7 Table**). Of these, 8 of 11 households had the same pathogen detected in multiple members within a two-week period, and 4 of 11 households had a pathogen detected in multiple members on the same day.

The prevalence of acute viral pathogen detection was comparable across malaria infection statuses: qPCR negative (40% with at least one respiratory or gastrointestinal viral infection), qPCR positive (32%), mNGS negative (38%), mNGS positive (40%), negative on both assays (40%), and positive on both assays (36%) (Tables 3 and S8).

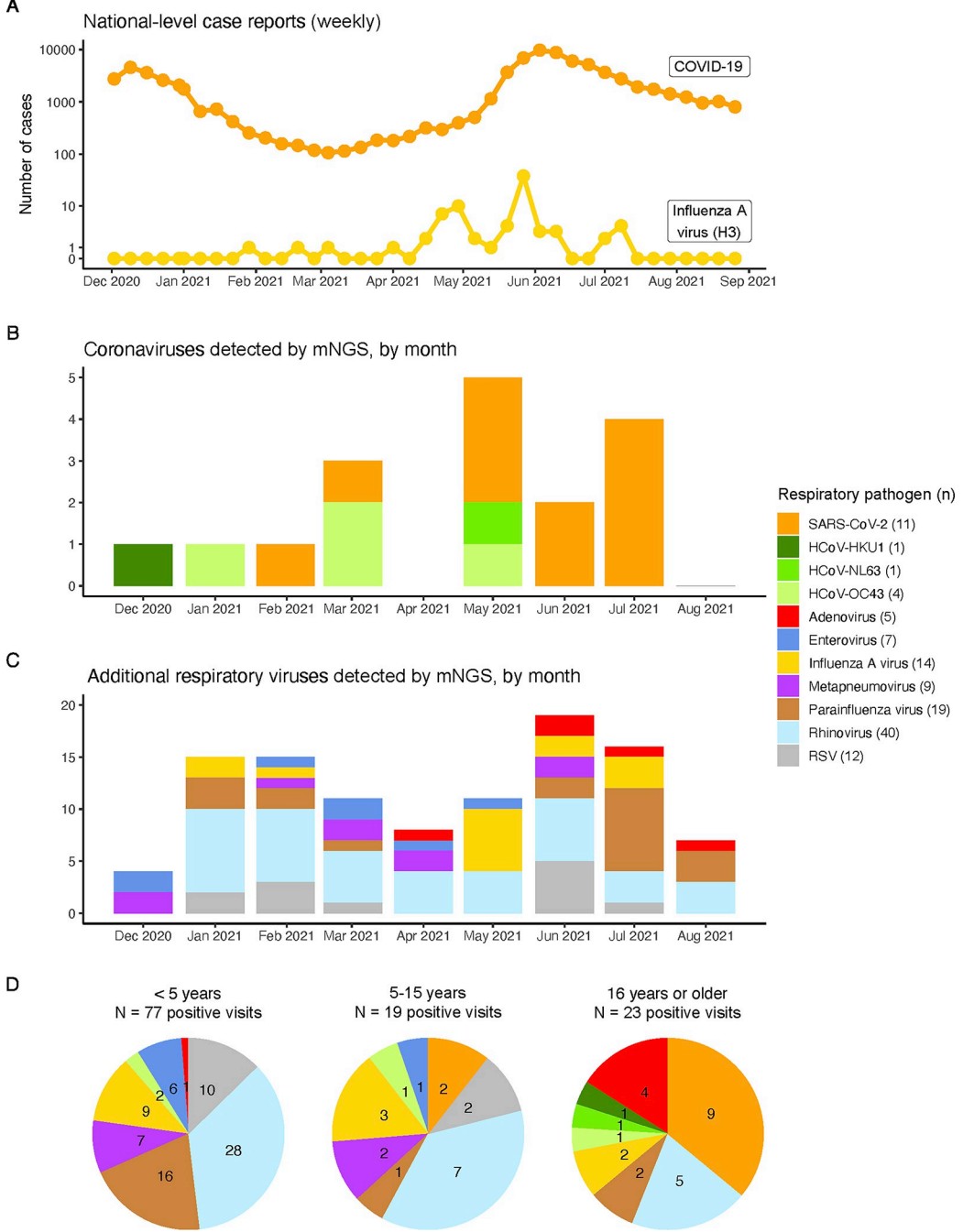

**Fig 3. Respiratory viral pathogens by detection month and age group.** If a virus was detected in both sample types from a given visit, it was counted only once. See legend for number of times each virus was detected in this study. **(A)** Weekly reported case counts of COVID-19 and Influenza A virus (H3 subtypes) in Uganda between December 2, 2020 and August 26, 2021. COVID-19 case counts downloaded from the Ugandan Ministry of Health COVID-19 Response Info Hub database [41]. Influenza case counts shown are the number of specimens positive for Influenza A virus (H3 subtype), downloaded from the World Health Organization's Global Influenza Surveillance and Response System (GISRS) database [42]. **(B)** Number of human coronaviruses identified. **(C)** Number of additional respiratory viruses identified. Note the differing y-axes in panels B and C. **(D)** Identified pathogens by participant age group, with the same color scale as in previous panels. RSV: Respiratory syncytial virus.

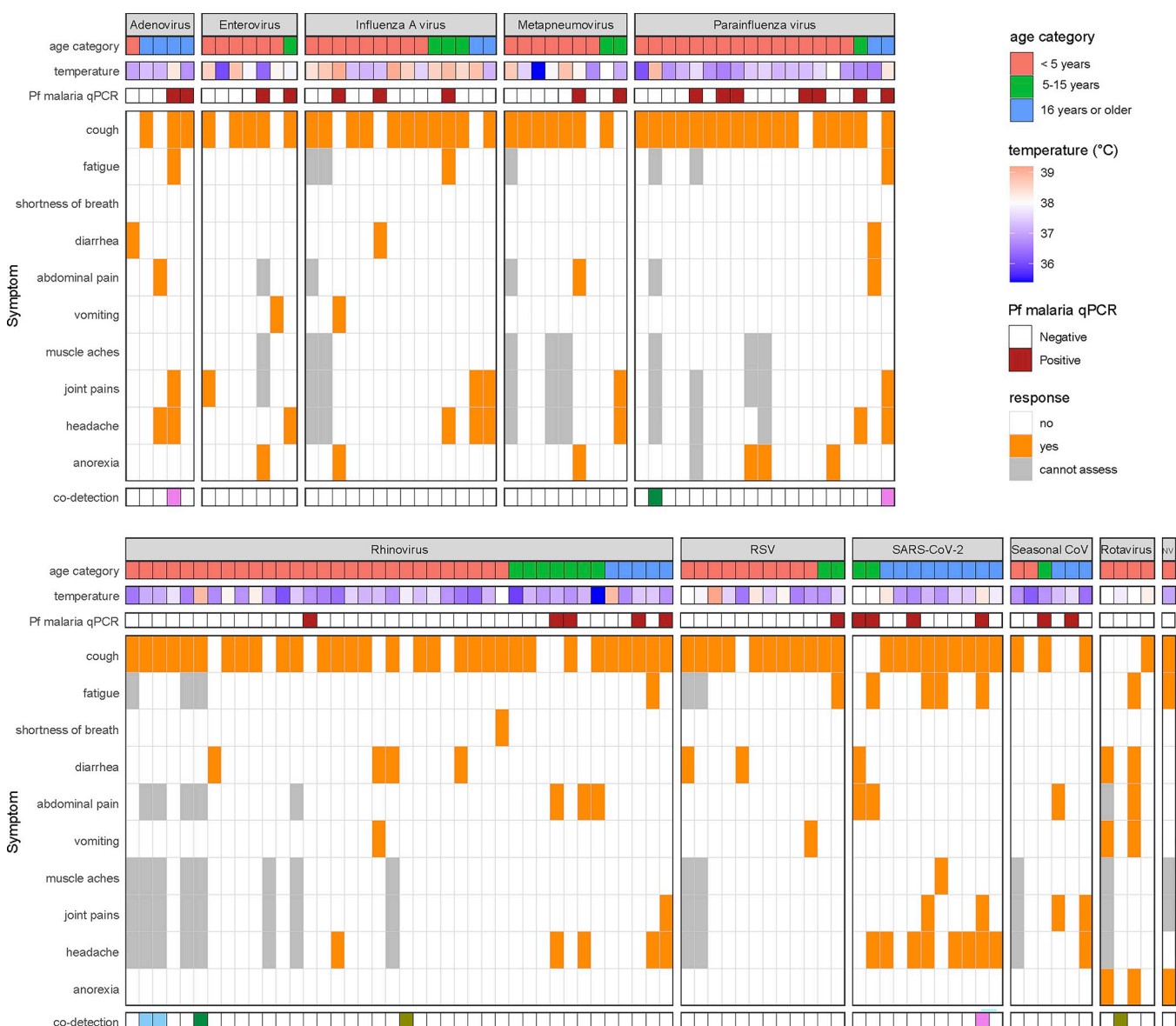

**Fig 4. Participant age, objective temperature, *Plasmodium falciparum* malaria qPCR result, and reported symptoms at each visit where a respiratory or gastrointestinal viral pathogen was detected.** Each column corresponds to a visit. See S3 Table for mapping column names (categories) to viral species. Multiple viral pathogens co-detected at the same visit are depicted by matching color cells in the bottom row. NV (last column): Norovirus.

## Genomic characterization of SARS-CoV-2, Influenza A virus, RSV

Of the 11 study visits where SARS-CoV-2 was detected by mNGS, 9 full genomes were recovered. The 9 full genomes represented multiple variant lineages, including Delta (5), Eta (3), and Alpha (1). The two partial genomes were assigned to the Delta variant lineage as well, though they were not included in downstream phylogenetic analyses. We performed phylogenetic inference of the 9 full genomes along with other recent SARS-CoV-2 genomes obtained from GISAID from the African region and globally (**Figs 5** and **S8A**). A description of the inferred phylogeny is provided in **S1 Text**. We found that the 9 genomes generated as part of this study clustered amongst the viral diversity observed across other Ugandan and African SARS-CoV-2 consensus genomes. The degree of divergence observed between our sequences

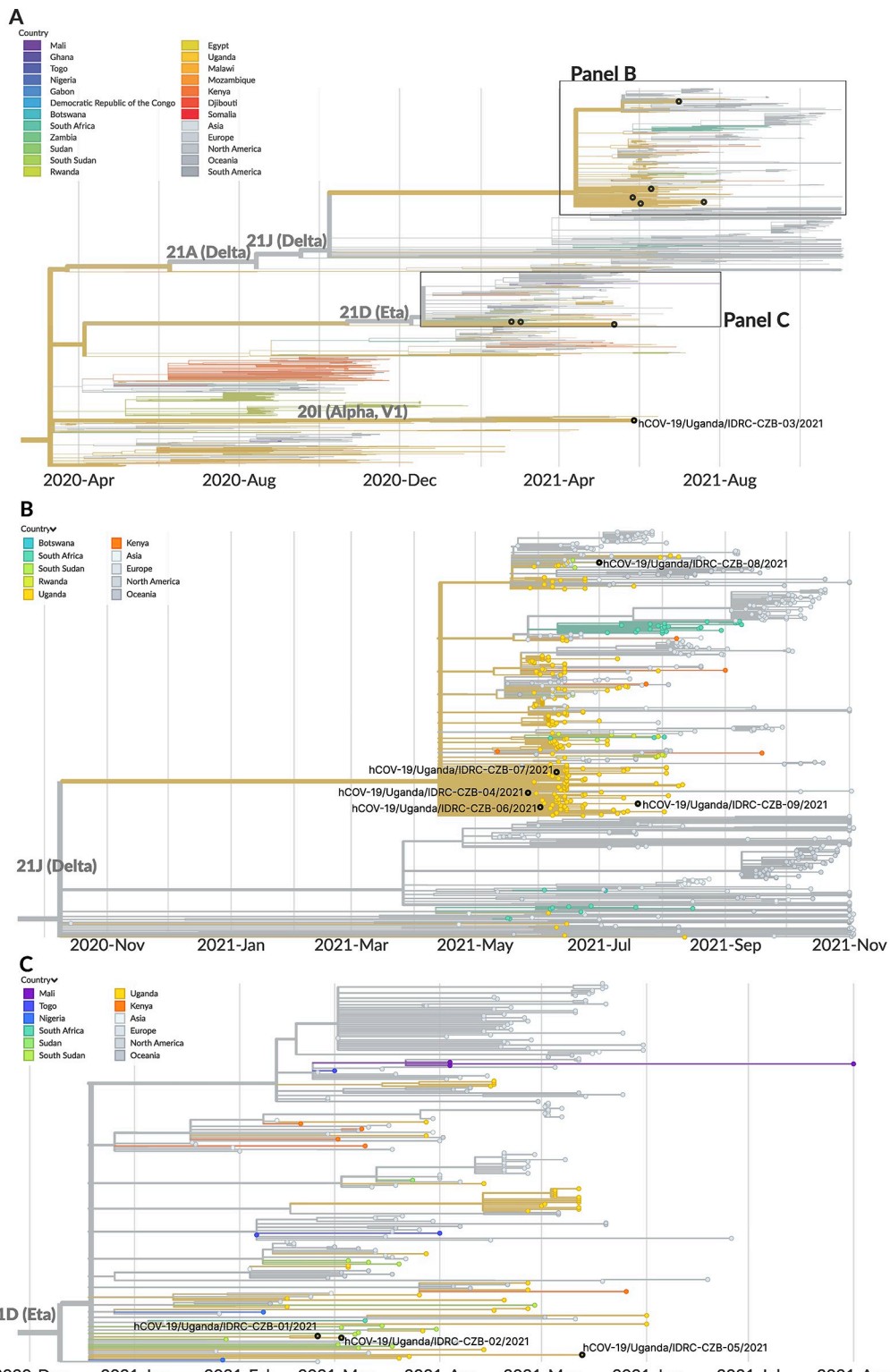

**Fig 5. Phylogeographic analysis of SARS-CoV-2 genomes generated in this mNGS study. (A)** Temporally-resolved phylogeny of SARS-CoV-2 genomes based on 2,677 SARS-CoV-2 genomes from the GISAID database closely related to the 9 sequences determined from this study. The 9 sequences determined from this study are shown in black overlaid circles. **(B)** Phylogeny narrowed in on the Delta clade (including 5 genomes from the study). Four of the Delta-lineage viruses fell within a polytomous clade defined by a C10977T mutation. The fifth virus grouped within a different clade

of Delta-lineage viruses defined by a G19117T mutation. (**C**) Phylogeny narrowed in on the Eta clade (including 3 genomes from the study). Two viruses grouped together, sharing C4570A, C13536T, C21811T mutations, and were separated by a C21846T mutation that was unique to hCoV-19/Uganda/IDRC-CZB-01/2021. In all panels, tip colors indicate country of origin (legend is shared by panels). Proximity-based subsampling was used on the focal set (i.e., the 9 genomes generated from this study and all Ugandan genomes from GISAID), grouped by year and month, with a maximum of 2,000 sequences.

and other publicly available SARS-CoV-2 consensus genomes was in line with the relatively low density of sampling in the region compared to other locations, implying that lineages likely circulated for longer periods before being sampled via genomic surveillance.

Of the 14 study visits where Influenza A virus (all H3N2 subtype) was detected by mNGS, 9 partial HA gene segments were recovered. We used these 9 HA segments, along with other recent Influenza A H3N2 HA segments obtained from GISAID from the African region and globally, to build a phylogenetic tree (**Figs 6** and **S8B**). A description of the inferred phylogeny is provided in **S1 Text**. We found that the HA sequences generated in this study clustered amongst the viral diversity sampled from Kenya, Zambia, the Democratic Republic of Congo, Mozambique, and South Africa. All 9 HA sequences generated in this study were grouped together in a single clade. The most recent common ancestor of the HA sequences generated as part of this study was inferred to circulate in November 2020 (95% confidence interval (CI): October 2020 to November 2020). The shape of our inferred tree was consistent with the 'ladder-like phylogeny' associated with continual immune selection that has previously been described for the HA gene of Influenza A H3N2 [43,44].

Of the 12 study visits where RSV was detected by mNGS, 9 full genomes were recovered. The 9 full genomes and 2 additional partial genomes were of the RSV-A subtype, with one partial genome of the RSV-B subtype. We used the 9 full genomes, along with other recent RSV genomes obtained from GISAID from the African region and globally, to build a phylogenetic tree (**Figs 7** and **S8C**). A description of the inferred phylogeny is provided in **S1 Text**. The RSV sequences generated as part of this study fell into 3 distinct clades. The inferred date for the most recent common ancestor of the RSV sequences generated as part of this study was September 2012 (95% CI: August 2010 to July 2013). This diversity among the RSV samples, illustrated through the lack of a common ancestor until almost 10 years prior to sampling, is likely explained by the very low density of RSV sampling in the region.

## Discussion

Here, we performed a comprehensive and systematic study of over 300 non-malarial febrile illnesses in a representative cohort from eastern Uganda, using mNGS on plasma and respiratory swabs collected between December 2020 to August 2021. We detected a viral pathogen in half of the illnesses occurring in young children, decreasing with age to just over one quarter for adults, for an average of 39% of illnesses overall. We identified the co-circulation of several important respiratory pathogens during this time period, including SARS-CoV-2, Influenza A (H3N2 subtype), and RSV. The composition of respiratory viral pathogens that were detected varied considerably by age, and to a lesser extent, by time. While respiratory viruses were primarily detected in respiratory swab samples as expected, there were instances in which respiratory and GI viruses were also (or only) detected in plasma samples, which could occur if a localized infection becomes systemic. In addition to detecting the potential causes of a number of illnesses, we were able to obtain consensus genomes from mNGS data for SARS-CoV-2, Influenza A, and RSV, and to relate them to other publicly available genomes from the region and globally.

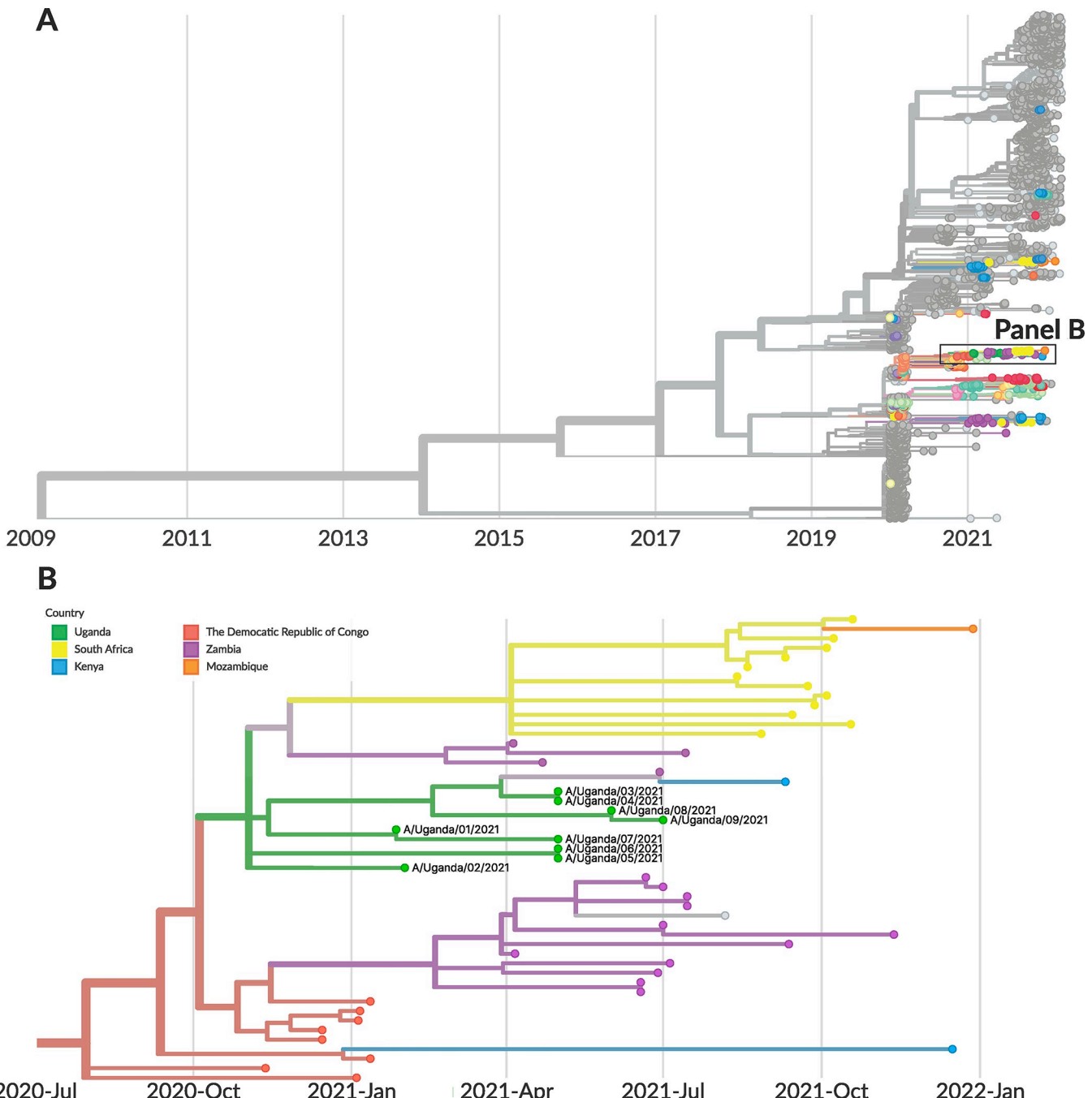

**Fig 6. Phylogeographic analysis of Influenza A (H3N2) genomes generated in this mNGS study.** (A) Temporally-resolved phylogeny of HA genes of Influenza A (H3N2) based on 3,816 Influenza A (H3N2) HA genes from the GISAID database and the 9 HA genes generated from this study. Points in color represent sequences from Africa, and points in greyscale represent sequences from other parts of the world. (B) Phylogeny narrowed in on the samples generated from this study (9 HA genes) in green. All of the HA sequences generated in this study were clustered in a clade defined by C1577T. All samples labeled as "Uganda" are from this study. Sub-sampling was used to get 40 sequences per country per year and month, apart from genomes originating from Africa, which were all included.

A number of epidemiologic and genomic trends emerge from this well-characterized data set. The timing of this study (i.e., during the first 8 months of 2021) coincides with the implementation of social distancing measures as a result of the global SARS-CoV-2 pandemic.

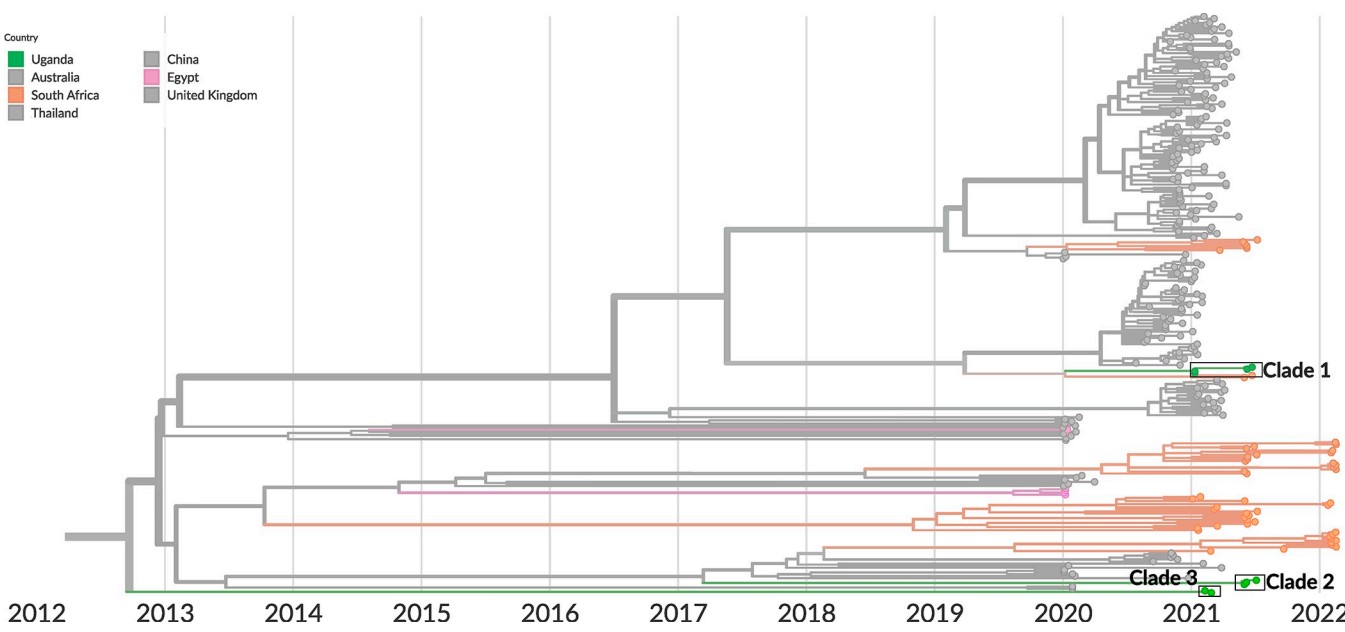

**Fig 7. Phylogeographic analysis of RSV genomes generated in this mNGS study.** Temporally-resolved phylogeny of the RSV genomes generated in this study, based on 460 RSV genomes from the GISAID database and the 9 RSV genomes generated from this study. All samples labeled as "Uganda" are from this study. The tree shows only sequences of the RSV-A subtype; none of the 9 RSV genomes generated in this study were of the RSV-B subtype. Tip colors indicate country of origin: Those in color represent sequences from Africa, and those in grayscale represent sequences from other parts of the world. Sub-sampling was used to get 40 sequences per country per year and month, apart from genomes originating from Africa, which were all included. Clade 1 is defined by 7 mutations (T2156C, C5008T, C7184T, G11704A, T1736C, C2129T, T7977C) from the closest basal virus on the tree. Clade 2 is defined by 59 mutations from the closest basal virus on the tree. Clade 3 is defined by 102 mutations from the closest basal virus on the tree.

Reductions in incidence, as well as shifts in timing, have been reported worldwide during the pandemic for other respiratory viruses such as influenza and RSV [45–49]. Resurgences following the subsequent lifting of social distancing measures have also been documented, driven by age-dependent patterns of immunity to different pathogens in a population [50–53]. Here, we identified the co-circulation of multiple respiratory pathogens in this cohort, including SARS-CoV-2, Influenza A, and RSV. Interestingly, the number of detections of Influenza A and RSV were larger than the number of detections of SARS-CoV-2 during this study's time period. While respiratory viral surveillance is limited in this setting, our findings on the temporal trends of Influenza A in this cohort are broadly consistent with data from the WHO Global Influenza Surveillance and Response System [42], which detected the H3 subtype of Influenza A circulating in Uganda during this time period. Direct comparison of data from the SARS-CoV-2 and influenza public health surveillance systems suggest that the case counts of these infections differ by several orders of magnitude. However, the systematic data from this study, while limited in scope (i.e., our 11 SARS-CoV-2 and 14 Influenza A detections), suggest that the relative numbers of cases may be much more similar. The degree of divergence observed between the SARS-CoV-2, Influenza A virus, and RSV sequences generated as part of this study and other publicly available sequences was in line with the relatively low density of sampling in the region. Lineages likely circulated for longer periods before being sampled via genomic surveillance. In particular, our ability to interpret the high diversity within our RSV sequences was limited by the sparsity of contextual sequences available.

We were also able to compare the detection of sub-microscopic malaria by qPCR and by mNGS. *Plasmodium falciparum* was detected in 57 plasma samples by mNGS, of which only 22 were previously characterized as positive by qPCR. Among samples with a non-zero value

on both assays, we found a positive trend between qPCR parasite densities and mNGS NT rPMs, and samples with NT rPMs above a threshold value were all positive by qPCR. These findings suggest that, even for these malaria infections that we know *a priori* to have low parasite densities, the results from this unbiased, pan-pathogen mNGS approach are correlated with results from a targeted, pathogen-specific assay. However, there were discrepancies between the qPCR and mNGS results in both directions, which is not unexpected since qPCR analyzed extracted DNA (in whole blood) while our mNGS strategy targeted RNA (in plasma), and these may have different abundances in biological samples.

There are a number of important caveats associated with the design of this epidemiologic study. First, the temporal window of sampling was limited and took place during a pandemic. As a result, these findings may not be representative of the potential causes of non-malarial fevers outside of the pandemic. Second, we only collected plasma and respiratory swabs, so are likely to have missed other potential causes of fevers that require additional specimen types to detect (i.e., stool samples for GI pathogens). Third, as we only collected samples from malaria blood smear-negative visits, we did not have the opportunity to test at febrile, malaria blood smear-positive visits for the presence of other pathogens causatively associated with fever. These limitations can theoretically be overcome with broader sample collection efforts. In addition, milder respiratory viral infections that were detected (e.g., rhinoviruses) may not be the etiology of the febrile illness. Lastly, we were unable to estimate the contribution of bacterial microbes to non-malarial fevers due to the difficulty of distinguishing bacterial microbes that were pathogenic from those that were commensal or contaminants. To address this key issue, we have started collecting convalescent samples from participants to establish potential background models for commensal microbes as an effort to distinguish pathogenic bacteria microbes. Important avenues of this future investigation include characterizing the microbiome and the prevalence of bacterial infections, investigating viral-bacterial interactions [54,55], and performing surveillance for antimicrobial resistance genes [56].

We emphasize that mNGS as applied here should be considered a research tool rather than a diagnostic, and thus would need to be validated with PCR or other clinically-approved assays to be used as the latter. As such, another important limitation is the unknown sensitivity of this mNGS assay to detect infection, which may vary by pathogen [57] and poses a challenge in analyzing the absolute and relative incidence of different pathogens. A potentially important factor affecting the sensitivity is that RNA sequencing depends on the quality and concentration of RNA, as this impacts the ability to capture all the sequences. However, positive correlations between NT rPMs from mNGS (which report the relative abundance of sequencing reads mapping to a specific microbe in a sample) and "gold standard" measurements (i.e., viral loads for SARS-CoV-2 [12] and other respiratory viruses [58], or qPCR parasite densities for malaria as presented in this work) suggest that thresholds can nonetheless be identified for specific pathogens. This would require dedicated studies testing with well-characterized targeted assays to establish the sensitivity of mNGS to detect a particular pathogen of interest which, while not trivial, would be straightforward and greatly improve the interpretability of these data. Additional data providing evidence for causality, such as including host transcriptomics (Rao et al. 2022), could also improve interpretability of mNGS results.

More broadly, this analysis underscores the utility and potential of multiplexed pathogen detection as a tool for surveillance and for better understanding the overall burden of different pathogens in a population (e.g., mNGS, the BioFire® RP2.1 Panel [59], or other multiplexed viral detection assays [60], as well as the potential complementary role of multiplexed pathogen serologic surveillance [61,62]). Cost is an important factor in the consideration of scaling up approaches such as this; there is a trade-off between sensitivity and cost, and these trade-offs will continue to evolve as molecular and informatic techniques improve. Studies such as ours

will lay the foundation for developing targeted approaches that can more sensitively measure pathogen burden, which can then be correlated with disease state. In addition, the ability to generate whole viral genomes through mNGS could be leveraged to fill in existing gaps in pathogen genomic data from resource-limited settings for important pathogens. Continuing to refine our understanding of the pathogen landscape of non-malarial febrile illness [4,63] in different settings and age groups could open the way to inform control interventions and case management guidelines, allow for the implementation and design of new rapid, low-cost diagnostics to inform clinical decision-making, and improve public health surveillance systems.

## Supporting information

**S1 Checklist. PLOS questionnaire on inclusivity in global research.**
(DOCX)

**S1 Fig. Distribution of sample input RNA mass and total reads by sample type.** (A) Input RNA in picograms. Blue line indicates 25 pg (i.e., amount of spike-in control in each sample). (B) Total reads per sample. These figures include the 292 plasma samples and 294 swab samples which passed CZ ID's QC filters and were thus included in this analysis. Sample input is calculated as: (25 pg / ERCC reads) * (total reads—ERCC reads).
(PDF)

**S2 Fig. Flowchart of PRISM Border Cohort study visits and corresponding mNGS sample collection conducted in this pilot.** The number of visits with mNGS sample collection among the 212 study participants is provided in the fourth row of Table 1.
(PDF)

**S3 Fig. Comparison of *Plasmodium falciparum* malaria qPCR parasite densities vs. mNGS NT rPMs in plasma samples.** The filled points (which have non-zero values for both qPCR and mNGS NT rPM to *Plasmodium falciparum*) were included in a linear regression of log parasite density (y-axis) versus log NT rPM (x-axis). Note that the model includes samples that did not meet our threshold criteria to be called as positive for *Plasmodium falciparum* by mNGS (indicated by color). The open points (which have a zero value by either or both assays, and have pseudocounts added for visualization purposes) were excluded from the regression. The horizontal line indicates the threshold to be called positive for *Plasmodium falciparum* by qPCR. Estimated slope = 1.01 and adjusted $R^2$ = 0.49.
(PNG)

**S4 Fig. Composition of microbial detections in plasma samples.** Each column is a distinct sample. The top row shows the sample input. The 25 pg threshold is shown by the dotted red line. The y-axes of this row vary by panel. The middle row represents the proportion of reads within the sample that corresponded to each kingdom prior to filtering. The bottom row represents the proportion of reads within the sample after filtering. Only filtered reads that map to bacterial, viral, or eukaryotic species are included in filtered barplots. Panel **(A)** includes samples in the bottom half of sample inputs, and panel **(B)** includes samples in the top half of sample inputs.
(PDF)

**S5 Fig. Composition of microbial detections in swab samples.** Each column is a distinct sample. The top row shows the sample input. The 25 pg threshold is shown by the dotted red line. The 2 samples in green denote failed ERCC, and thus not the accurate sample input. The y-axes of this row vary by panel. The middle row represents the proportion of reads within the sample that corresponded to each kingdom prior to filtering. The bottom row represents the

proportion of reads within the sample after filtering. Only filtered reads that map to bacterial, viral, or eukaryotic species are included in filtered barplots. Panel (**A**) includes samples in the bottom half of sample inputs, and panel (**B**) includes samples in the top half of sample inputs.
(PDF)

**S6 Fig. Comparison of viral microbes detected in (paired) plasma and swab samples.** All viral microbe species from Fig 2E are included; respiratory and gastrointestinal viral microbes that are known to be pathogenic to humans (see S3 Table) are highlighted with an asterisk.
(PDF)

**S7 Fig. Respiratory pathogen-specific prevalence among the sample population by age group.** Prevalence was estimated as the probability of detection by mNGS. We assumed that each pathogen detection was independent due to the infrequency of co-detections of respiratory viruses in this study (3 of 119 visits). The point depicts the posterior median probability and the outer interval is the 95% credible interval, using a binomial model.
(PDF)

**S8 Fig. Analysis of SARS-CoV-2, Influenza A, and RSV sequences generated in this mNGS study, with respect to nucleotide divergence. (A)** 9 SARS-CoV-2 genomes from this study, with differences in the number of mutations on the x-axis. **(B)** 9 Influenza A (H3N2) HA gene segments from this study, with divergence (number of mutations per site) on the y-axis. **(C)** 9 RSV genomes from this study, with divergence (number of mutations per site) on the y-axis. Samples from the same household are shown in the same color (the samples labeled in black are from a household with a singleton sample).
(PDF)

**S1 Table. Protocol details for the two processing/sequencing batches.**
(PDF)

**S2 Table. Accession numbers for SARS-CoV-2, Influenza A virus, and RSV sequences.**
(PDF)

**S3 Table. Classification of viral microbes/pathogens detected.**
(PDF)

**S4 Table. Frequency table of binary results for *Plasmodium falciparum* malaria by mNGS and qPCR for the 292 plasma samples tested by mNGS.**
(PDF)

**S5 Table. 132 bacterial genera detected in plasma samples.**
(PDF)

**S6 Table. 289 bacterial genera detected in swab samples.**
(PDF)

**S7 Table. Household co-detection of viral pathogens, and time between infections in a household.** Days between infections in the 11 households that had multiple individuals with the same respiratory viral pathogen detected during the sampling period. HH: household.
(PDF)

**S8 Table. Co-detections of malaria in plasma (by qPCR and/or by mNGS) and selected viral pathogenic microbes detected in plasma/swab (by mNGS) at a given visit.** These are for the 273 visits that had paired mNGS sample collections, as summarized in Table 3. The included viruses are those that are designated as "respiratory" or "gastrointestinal" pathogens

in S3 Table.
(PDF)

**S1 Text. Supplementary results.**
(DOCX)

**S1 File. Clinical and demographic characteristics of the study participants.**
(CSV)

## Acknowledgments

We thank all study participants who took part in this study. We also thank the clinical, laboratory, and data study team and the Infectious Diseases Research Collaboration for administrative and technical support. We acknowledge Maíra Phelps from the Chan Zuckerberg Biohub for assistance with sample coordination, Dan Lu from Chan Zuckerberg Initiative for assistance with the GISAID upload, and the Genomic Team at the Chan Zuckerberg Biohub for assistance with sequencing: Norma Neff, Angela Detweiler, Honey Mekonen and Sheryl Paul.

## Author Contributions

**Conceptualization:** Vida Ahyong, Isabel Rodríguez-Barraquer, Cristina M. Tato, Bryan Greenhouse.

**Data curation:** Lusajo Mwakibete, Vida Ahyong.

**Formal analysis:** Lusajo Mwakibete, Saki Takahashi, Vida Ahyong, Allison Black, Isabel Rodríguez-Barraquer, Cristina M. Tato, Bryan Greenhouse.

**Funding acquisition:** Prasanna Jagannathan, Isabel Rodríguez-Barraquer, Bryan Greenhouse.

**Investigation:** Lusajo Mwakibete, Saki Takahashi, Vida Ahyong, Allison Black, John Rek, Isaac Ssewanyana, Moses Kamya, Grant Dorsey, Prasanna Jagannathan, Isabel Rodríguez-Barraquer, Cristina M. Tato, Bryan Greenhouse.

**Methodology:** Lusajo Mwakibete, Saki Takahashi, Vida Ahyong, Allison Black, Prasanna Jagannathan, Isabel Rodríguez-Barraquer, Cristina M. Tato, Bryan Greenhouse.

**Project administration:** John Rek, Isaac Ssewanyana, Moses Kamya, Grant Dorsey, Isabel Rodríguez-Barraquer, Cristina M. Tato, Bryan Greenhouse.

**Resources:** Vida Ahyong, Grant Dorsey, Prasanna Jagannathan, Isabel Rodríguez-Barraquer, Cristina M. Tato, Bryan Greenhouse.

**Software:** Vida Ahyong.

**Supervision:** Saki Takahashi, Vida Ahyong, Isabel Rodríguez-Barraquer, Cristina M. Tato, Bryan Greenhouse.

**Visualization:** Lusajo Mwakibete, Saki Takahashi, Allison Black.

**Writing – original draft:** Lusajo Mwakibete, Saki Takahashi, Vida Ahyong, Allison Black, John Rek, Isaac Ssewanyana, Moses Kamya, Grant Dorsey, Prasanna Jagannathan, Isabel Rodríguez-Barraquer, Cristina M. Tato, Bryan Greenhouse.

**Writing – review & editing:** Lusajo Mwakibete, Saki Takahashi, Vida Ahyong, Allison Black, John Rek, Isaac Ssewanyana, Moses Kamya, Grant Dorsey, Prasanna Jagannathan, Isabel Rodríguez-Barraquer, Cristina M. Tato, Bryan Greenhouse.

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
