## [Decision Letter · Decision Letter 0]

13 Dec 2022

PGPH-D-22-01529

Metagenomic next-generation sequencing to characterize etiologies of non-malarial fever in a cohort living in a high malaria burden area of Uganda

Dear Dr. Takahashi,

Thank you for submitting your manuscript to PLOS Global Public Health. After careful consideration, we feel that it has merit but does not fully meet PLOS Global Public Health’s publication criteria as it currently stands. Therefore, we invite you to submit a revised version of the manuscript that addresses the points raised during the review process.

We look forward to receiving your revised manuscript.

Kind regards,

Raquel Muñiz-Salazar, Ph.D.

Academic Editor

Journal Requirements:

2. In the online submission form, you indicated that your data will be submitted to a repository upon acceptance.  We strongly recommend all authors deposit their data before acceptance, as the process can be lengthy and hold up publication timelines. Please note that, though access restrictions are acceptable now, your entire data will need to be made freely accessible if your manuscript is accepted for publication. This policy applies to all data except where public deposition would breach compliance with the protocol approved by your research ethics board. If you are unable to adhere to our open data policy, please kindly revise your statement to explain your reasoning and we will seek the editor's input on an exemption. Please be assured that, once you have provided your new statement, the assessment of your exemption will not hold up the peer review process.

Additional Editor Comments (if provided):

The approach is interesting, however, the autorhs have to review the discussion and conclusions following the reviewer´s comments.

Reviewers' comments:

Reviewer's Responses to Questions

**Comments to the Author**

1. Does this manuscript meet PLOS Global Public Health’s publication criteria? Is the manuscript technically sound, and do the data support the conclusions? The manuscript must describe methodologically and ethically rigorous research with conclusions that are appropriately drawn based on the data presented.

Reviewer #1: Yes

Reviewer #2: Partly

Reviewer #3: Yes

2. Has the statistical analysis been performed appropriately and rigorously?

Reviewer #1: I don't know

Reviewer #2: I don't know

Reviewer #3: Yes

3. Have the authors made all data underlying the findings in their manuscript fully available (please refer to the Data Availability Statement at the start of the manuscript PDF file)?

Reviewer #1: Yes

Reviewer #2: Yes

Reviewer #3: Yes

4. Is the manuscript presented in an intelligible fashion and written in standard English?

Reviewer #1: Yes

Reviewer #2: Yes

Reviewer #3: Yes

5. Review Comments to the Author

Reviewer #1: The authors here present the application of metagenomic next-generation sequencing (mNGS) for broad genomic-level detection of infectious agents to elucidate the potential causes of non-malarial fevers. Based on their data, they identified a viral pathogen in half of the illnesses in young children and in 40% illness overall. The study seems well-designed, however, the authors are advised to compare their observations with other mNGS datasets to evaluate overlapping potential pathogenic species that might not contribute to fever. The authors are also advised to mention the ICD code for fever as applied in this study. Other specific points that are recommended are as follows:

1. Do the authors think that the vaccination status of the cohort might impact the readout?

2. R codes may be shared for data reproducibility through platforms like GitHub as the methodology needs to be provided at the highest possible resolution. The authors may also like to mention if the analysis done in R is part of the snakefile as it is not mentioned in the manuscript.

3. Line 291 (Results), the number of plasma and swab sample don’t match to Figure S1 or Table S4. May please correct.

4. 212 individuals should be represented in Figure S2 for better clarity of samples collected.

5. Line 311 (Results), the text says ‘additional 35 visits’, however, Table 3 says 34, may please correct.

6. Line 329 (Results), it says 468 bacteria, however, total from Table S5 (132) and Table S6 (289) is 421, please check.

7. Lines 480-481 (Discussion), may please elaborate on the statement of why the authors think that the relative number of cases may be much more similar?

8. Data access from http://czid.org/pub/7pxmXbpFTL is not working even after several attempts.

Reviewer #2: Mwakibete et al. applied mNGS for the epidemiological characterization of respiratory and plasma samples from febrile patients without microscopically diagnosed malaria in Uganda. The approach is interesting and educative for readers, however, I do not agree with all interpretations and conclusions of the researchers.

1.) In the first paragraph of their introduction, the authors thoroughly explain the phenomenon of semi-immunity after multiple exposition to pathogenic Plasmodium spp. However, they should also stress the difficulty of discriminating clinically apparent infections from harmless colonization in case of the most facultatively pathogenic microorganisms if they are detected at primarily non-sterile sampling sites (like, e.g., many human mucous membranes).

2.) Materials and methods, sub-heading “study population and design”) The data would be better interpretable if the authors presented diagnostic accuracy values (sensitivity, specificity, limit-of-detection) for the applied reference testing approaches like the mentioned “ultrasensitive qPCR for Plasmodium falciparum malaria”. This also applies for potentially undesired side effects on target nucleic acid sequences associated with the depletion of human RNA (page 7, line 149).

3.) The authors should at least comment in the discussion on the diagnostic accuracy and interpretability of CZ ID-based sequence assignments. If the assignments are based on comparisons with deposited sequences at NCBI’s nucleotide, false sequence deposits will most likely influence the reliability of the assignments.

4.) Materials and methods, sub-heading “mNGS analysis using CZ ID”, lines 170-172: The chosen general exclusion of bacterial reads considerably limits the diagnostic value for infections with expected low pathogen loads, for example, in case of bacteremia / sepsis. The authors should discuss how this limitation might be overcome by future mNGS approaches.

5.) Discussion, first paragraph: Similar as already stated in my first comment, I have serious doubt whether the detected respiratory viruses have always been the reasons for the recorded febrile illnesses, considering the high variance of infection-associated symptoms. The authors should discuss this issue more critical. From my point of view, I don’t really see why viral detections should generally be more etiologically relevant than detections of facultatively pathogenic bacteria.

6.) Discussion, lines 512-513 “background microbiome models”: The authors should discuss the interference of such background subtraction with the detection of infections associated with low pathogen loads. Will there be a solution for this diagnostic dilemma?

7.) Discussion, lines 526-528: This comment by the authors strongly supports my point two; so, please provide the respective information (as far as it is available).

8. Discussion/conclusions, lines 530-534: I do not fully agree with this conclusion. From my point of view, the authors’ work clearly shows the obstacles to be overcome to reliably interpret such multiplexed data rather than providing solutions regarding the interpretation of the diagnostic results. Future research will have to focus on approaches of reliable assignment of etiological relevance to the findings of multiplexed diagnostic assessments in order to guide the clinicians’ decisions. Not to forget the influence of Bayes’ theorem in situations of hypothesis-free testing associated with low pretest-probability. Such questions will require proper answers before mNGS can be reliably used for diagnostic purposes in the future.

Reviewer #3: This manuscript describes a RNA-based metagenomic next-gen sequencing (mNGS) analysis of approximately 300 plasma and nasal swab samples from a cohort study in Uganda conducted in 2020/2021, with the general aim of elucidating the causal agent of non-malarial fevers in the region. The dataset is thoroughly described, and the analyses appear to be competently performed, with the CZ ID platform doing much of the mNGS analysis. This work will be of broad interest given it addresses the question of how informative metagenomic sequencing can potentially be for clinical diagnosis and multiplexed genomic epidemiology of infectious diseases. Most importantly, though this manuscript very responsibly addresses some limitations of the study, it does not provide a map for what would be required to use such data to infer causality-how much data would be required, from how many subjects, and what sample types, and in what range of locations, to build some kind of general causal classifier of infectious agents from metagenomic sequencing data? Does this study suggest that is a realistic endeavor?

The manuscript could be improved by addressing/clarifying the following generally minor issues:

1) Line 80/81: The statement is made that RNA mNGS can quantitatively capture the relative abundance of different microbes/viruses in a clinical sample, and do so in a way that is superior to genomic DNA NGS metagenomic sequencing. There is no citation for this assertion and no validation of this statement in the results presented. I would imagine it could be tricky to evaluate relative abundance of various bacteria/eukaryotes with any precision without knowing more about the baseline transcriptional profiles of many organisms, library biases, etc. Perhaps restrict this statement to RNA viruses, or otherwise cite publications validating this.

2) Line 151: The sequencing approach is described generally, but it would be useful here to mention the target coverage (profiled in a supp fig, but most will not dig that up), along with a justification for a certain coverage. Coverage level (and cost) is a major factor in the consideration of scaling up approaches such as this, so it could be useful to address cost vs. information content as an issue in the Discussion.

3) Line 288: The authors note that 'a quarter' of subjects who were smear-negative for malaria at the time of sample collection were smear-positive at a subsequent visit. That seems like a lot, especially if qPCR was also missing some malaria infections. More precision here would be useful here in citing sample numbers; were subjects that subsequently became smear positive more likely to show Plasmodium reads in the mNGS analysis, even if negative by qPCR? This is the kind of issue that seems important to address if the community is ever going make mNGS data more interpretable with regard to causality.

4) Line 335: What does it mean when swab samples detect zero microbial species? Is that a technical assay failure?

5) Line 453: Detection of viral pathogens in clinical samples is key, but the authors should remind readers here or elsewhere in the Discussion that the ultimate goal would be to attribute symptoms to a particular detected pathogen. What would be required for that? Certainly lots of metagenomic profiles from healthy, asymptomatic individuals of various ages, as a start.

6. PLOS authors have the option to publish the peer review history of their article (what does this mean?). If published, this will include your full peer review and any attached files.

**Do you want your identity to be public for this peer review?** For information about this choice, including consent withdrawal, please see our Privacy Policy.

Reviewer #1: No

Reviewer #2: No

Reviewer #3: No

---

## [Decision Letter · Decision Letter 1]

13 Apr 2023

Metagenomic next-generation sequencing to characterize potential etiologies of non-malarial fever in a cohort living in a high malaria burden area of Uganda

PGPH-D-22-01529R1

Dear Dr Takahashi,

We are pleased to inform you that your manuscript 'Metagenomic next-generation sequencing to characterize potential etiologies of non-malarial fever in a cohort living in a high malaria burden area of Uganda' has been provisionally accepted for publication in PLOS Global Public Health.

Best regards,

Julia Robinson

Executive Editor

Reviewer Comments (if any, and for reference):

Reviewer's Responses to Questions

**Comments to the Author**

1. If the authors have adequately addressed your comments raised in a previous round of review and you feel that this manuscript is now acceptable for publication, you may indicate that here to bypass the “Comments to the Author” section, enter your conflict of interest statement in the “Confidential to Editor” section, and submit your "Accept" recommendation.

Reviewer #2: All comments have been addressed

Reviewer #3: All comments have been addressed

2. Does this manuscript meet PLOS Global Public Health’s publication criteria? Is the manuscript technically sound, and do the data support the conclusions? The manuscript must describe methodologically and ethically rigorous research with conclusions that are appropriately drawn based on the data presented.

Reviewer #2: Yes

Reviewer #3: Yes

3. Has the statistical analysis been performed appropriately and rigorously?

Reviewer #2: N/A

Reviewer #3: Yes

4. Have the authors made all data underlying the findings in their manuscript fully available (please refer to the Data Availability Statement at the start of the manuscript PDF file)?

Reviewer #2: Yes

Reviewer #3: Yes

5. Is the manuscript presented in an intelligible fashion and written in standard English?

Reviewer #2: Yes

Reviewer #3: Yes

6. Review Comments to the Author

Reviewer #2: My points have been addressed in an acceptable manner.

Reviewer #3: (No Response)

7. PLOS authors have the option to publish the peer review history of their article (what does this mean?). If published, this will include your full peer review and any attached files.

**Do you want your identity to be public for this peer review?** For information about this choice, including consent withdrawal, please see our Privacy Policy.

Reviewer #2: No

Reviewer #3: No
